



# Simulating the seeder-feeder impacts on cloud ice and precipitation over the Alps

Zane Dedekind[1,2,*], Ulrike Proske[2,*], Sylvaine Ferrachat[2], Ulrike Lohmann[2], and David Neubauer[2]

[1]Department of Earth, Ocean, and Atmospheric Sciences, University of British Columbia, Earth Sciences Building, 2207 Main Mall, Vancouver, BC, V6T 1Z4, Canada
[2]Institute of Atmospheric and Climate Science, ETH Zurich, Switzerland
[*]These authors contributed equally to this work.

**Correspondence:** Zane Dedekind (zane.dedekind@ubc.ca) and David Neubauer (david.neubauer@env.ethz.ch)

**Abstract.** The ice phase impacts many cloud properties as well as cloud lifetime. Ice particles that sediment into a lower cloud from an upper cloud (external seeder-feeder process) or into the mixed-phase region of a deep cloud from cirrus levels (internal seeder-feeder process) can influence the ice phase of the lower cloud, amplify cloud glaciation and enhance surface precipitation. Recently, numerical weather prediction modeling studies have aimed at representing the ice crystal number concentration in mixed-phase clouds more accurately by including secondary ice formation processes. The increase in the ice crystal number concentration can impact the number of ice particles that sediment into the lower cloud and alter its composition and precipitation formation. In the Swiss Alps, the orography permits the formation of orographic clouds, making it ideal for studying the occurrence of multi-layered clouds and the seeder-feeder process. We present results from a case study on May 18, 2016, showing the occurrence frequency of multi-layered clouds and the seeder-feeder process. About half of all observed clouds were categorized as multi-layered, and the external seeder-feeder process occurred in 10% of these clouds. In between cloud layers, $\approx 60\%$ of the ice particle mass was lost due to sublimation or melting. The external seeder-feeder process was found to be more important than the internal seeder-feeder process with regard to the impact on precipitation. In the case where the external seeder-feeder process was inhibited, the average surface precipitation and riming rate over the domain were both reduced by 8.5% and 3.9%, respectively. When ice-graupel collisions were allowed, further large reductions were seen in the liquid water fraction and riming rate. Inhibiting the internal seeder-feeder process enhanced the liquid water fraction by 6% compared to a reduction of 5.8% in the cloud condensate and, therefore, pointing towards the deamplification in cloud glaciation and a reduction in surface precipitation. Adding to the observational evidence of frequent seeder-feeder situations at least over Switzerland Proske et al. (2021), our study highlights the extensive influence of sedimenting ice particles on the properties of feeder clouds as well as on precipitation formation.





## 1 Introduction

Clouds are important for Earth's climate, modulating both the radiation balance and the water cycle. In particular, clouds' ice content determines many of their key properties such as their albedo and lifetime. Therefore, its correct representation in climate and numerical weather prediction models is imperative for improving weather forecasts and climate model representativeness.

In the atmosphere, ice forms homogeneously only at temperatures below $-35\,°C$. At mixed-phase temperatures ($0\,°C > T > -35\,°C$), ice forms heterogeneously on ice nucleating particles (INPs; e.g. Pruppacher and Klett (2010); Murray et al. (2010); Herbert et al. (2015); Kanji et al. (2017a) and references therein). In the absence of INPs, cloud droplets remain supercooled at mixed-phase temperatures. In a supercooled cloud, few ice crystals can exert a disproportionally large influence. They can multiply through secondary ice production (Korolev and Leisner, 2020). Several secondary ice production mecha-

nisms exist: the rime splintering process (also known as the Hallett-Mossop process, Hallett and Mossop, 1974; Mossop and Hallett, 1974), frozen droplet shattering (Lauber et al., 2018), or ice-ice collisional breakup (Vardiman, 1978; Takahashi et al., 1995). Ice crystals can grow by riming or vapor deposition (including the rapid growth via the Wegener-Bergeron-Findeisen process, where ice crystals grow at the expense of cloud droplets when the air is subsaturated with respect to liquid water but supersaturated with respect to ice (Wegener, 1911; Bergeron, 1935; Findeisen, 1938)). Eventually, the cloud will glaciate if the

ice crystals do not fall out as precipitation beforehand. Such effects can be triggered by ice crystals falling into a mixed-phase cloud (MPC) from above. Thereby, the higher cloud provides ice seeds, which feed on the moisture provided by the lower cloud and destabilize it as described above. This process is termed the seeder-feeder process.

     Originally, the seeder-feeder process was proposed to explain precipitation enhancement over mountains. Here, an orographically formed cloud acts as the feeder cloud, in which the sedimenting ice particles from the feeder cloud grow by accretion,

aggregation or riming after colliding with hydrometeors in the feeder cloud. Observations in field studies at various locations (Dore et al., 1999; Purdy et al., 2005; Hill et al., 2007) and idealized modeling studies (e.g. Carruthers and Choularton, 1983; Robichaud and Austin, 1988) have confirmed the seeder-feeder process. The seeder-feeder process for the ice phase was first proposed by Braham (1967). Here, ice particles fall into a lower cloud in the mixed-phase temperature regime, which does not need to be an orographic cloud. This ice-phase seeder-feeder process is the focus of the present study. In a wider sense, ice

crystals falling into a lower part of the same cloud can be understood as an internal seeder-feeder process (Hobbs et al., 1980).

     Natural cloud seeding by ice crystals has been inferred from remote sensing and observed during aircraft campaigns (Dennis, 1954; Hobbs et al., 1980, 1981; Locatelli et al., 1983; Hobbs et al., 2001; Pinto et al., 2001; Fleishauer et al., 2002; Ansmann et al., 2008; Creamean et al., 2013; Ramelli et al., 2020). It requires multi-layer clouds, in between which seeding ice crystals do not sublimate completely. The occurrence frequency of such situations in Svalbard has been estimated at 29 % during a

measurement campaign (including cloudy and non-cloudy days) by Vassel et al. (2019) using radiosonde and radar observations in combination with sublimation calculations. Seifert et al. (2009) and Ansmann et al. (2009) estimated that 10 % of ice-containing clouds measured by lidar over Leipzig at $-20\,°C$ were naturally seeded. Recently, a more thorough estimate has been derived using a 10-year lidar radar (DARDAR)-CLOUD satellite data set over Switzerland combined with sublimation calculations: Proske et al. (2021) found that external seeder-feeder situations occurred in 13 %, and internal seeder-feeder



situations in $18\,\%$ of the time in the observations. This presents a lower estimate since only ice clouds at $T<-35\,°\text{C}$ were considered as seeder clouds in their analysis. However, to estimate the importance of the seeder-feeder process, the occurrence frequency of seeding ice crystals reaching a lower cloud as seeds needs to be combined with an estimate of their impact in the feeder cloud.

Dietlicher et al. (2019) found that in the global climate model, ECHAM-HAM, ice crystal seeding is the most important ice
formation pathway. While this shows that seeding might have a large effect globally, we here study the effect on single clouds and precipitation in order to enhance our process understanding.

The impact of natural cloud seeding is difficult to observe in reality, as the enhancement from seeding needs to be separated from the precipitation that would also be occurring without any seeding. Still, e.g. Locatelli et al. (1983) found an intensification of the precipitation rate by 0.01 to $0.07\,\text{mm}\,\text{h}^{-1}$ through riming in the feeder cloud. This was corroborated by their observation
of seeding ice crystals growing by vapor deposition in the feeder cloud, where they found enhanced ice needle concentrations. In a study of artificial dry ice seeding in which unseeded and seeded clouds were compared, Hobbs et al. (1981) confirmed that seeding ice crystals changed the cloud drop size spectra and enhanced precipitation formation.

Such a comparison is easier to achieve in modeling studies, which have also been employed to study the impact of ice crystal seeding. Rutledge and Hobbs (1983a) simulated warm-frontal lifting. They found that the mass added to seeding ice crystals
in the feeder cloud made up 75% of the total precipitation mass. In the absence of seeding the hydrometeors remained as cloud droplets. In a simulated case study of an observed seeder-feeder episode on the Iberian Peninsula, Fernández-González et al. (2015) found that the seeding ice crystals collected supercooled cloud droplets in the feeder cloud, which led to moderate snowfall and prevented freezing drizzle. The seeder cloud also has radiative effects on the feeder cloud. Chen et al. (2020) used idealized model simulations to study this effect and noticed that the downwelling longwave radiation from the seeder cloud
reduced the cloud top radiative cooling from the feeder cloud. Together with the latent heat release in the feeder cloud from glaciation caused by the seeded ice particles, this led to the dissipation of the feeder cloud in their case.

While such idealized studies help to understand the processes involved, their use for a quantitative impact assessment is limited. Additionally, the sensitivity study approach of removing the seeding cloud entirely, applied in modeling studies up to date, possibly introduces large perturbations. In this study, we want to add a mechanistic understanding of the effect of natural
cloud seeding to the evidence for the frequent occurrence of natural cloud seeding over Switzerland (Proske et al., 2021). To this end, we simulate a seeder-feeder case study in the regional atmospheric model COSMO. The situation is over the Swiss Alps, where the topography permits orographic clouds to form, and where we can combine our results with the natural cloud seeding frequency estimates over Switzerland from Proske et al. (2021). Most importantly, we remove only those seeding ice particles that would reach a feeder cloud immediately above the feeder cloud, in order to reduce feedbacks. The following
section (Sect. 2) further explains the employed methods and model setup. In Sect. 3, results from the sensitivity studies of both the internal and external seeder feeder cases are presented and discussed. Conclusions and an outlook make up Sect. 4.



## 2    Methods

### 2.1    Model setup

To understand the effect of the internal and external seeder-feeder process on surface precipitation better, we use the non-
hydrostatic limited-area atmospheric model of the Consortium for Small Scale Modelling (COSMO version 5.4.1b;  Baldauf
et al., 2011). This COSMO version has recently been used to study wintertime orographic MPCs in the Swiss Alps (Lohmann
et al., 2016; Henneberg et al., 2017; Dedekind et al., 2021, 2023). We use a two-moment cloud microphysics scheme within
COSMO with six hydrometeor categories, including hail (Blahak, 2008), graupel, snow, ice crystals, raindrops and cloud
droplets (Seifert and Beheng, 2006). The two-moment cloud microphysics scheme includes the rime splintering process, which
is used in all simulations in this study. The output of ice, snow, graupel and hail precipitation fluxes was implemented, especially
for this study, to obtain these hydrometeors' sedimentation rates as needed for the analysis of the seeder-feeder process. Here
we defined two different scenarios for the seeder-feeder process; the external- and internal seeder-feeder process (Fig. 1).

The external seeder-feeder process is defined as when ice particles sediment into a separate lower cloud layer. To inhibit
it, in a sensitivity simulation (No-Ext-SF), we removed the seeding ice particles where the cloud area fraction (CLC, a full
3D function) is again larger than 0 in the level just above the feeder cloud to prevent any interaction with the feeder cloud.
At the level where the seeding ice particles are removed, they are not sublimated or melted, but rather they are artificially
removed. Therefore, no adiabatic cooling takes place in this layer for the No-Ext-SF simulation. In the No-Ext-SF simulation,
the increase in ice crystals can then only occur through primary or secondary ice formation processes in the feeder cloud. In
COSMO, ice crystals and snow are typically described as hexagonal plates and "mixed aggregates", respectively (Seifert and
Beheng, 2006,  and references therein). Additionally, an option is included in which the ice crystal and snow particle shapes
are changed to spheres without changing the mass of that particle (No-Ext-SF_Sph simulation) when they sediment out of
the seeder-cloud (Fig. 1). Spherical ice particles have a smaller ventilation coefficient for Reynolds numbers less than 100
and sublimate at a slower rate (Wang, 2013). In conjunction with the slower sublimation rate, they also have a smaller drag
coefficient and therefore fall further distances in dry air. Proske et al. (2021) conducted sublimation calculations of rosettes,
plates and spheres and showed that spheres led to successful seeding in 64% of the cases. This is 17% and 22% higher than
plates and rosettes, respectively. Therefore, we anticipate that spherical ice particles should enhance the seeder-feeder process
and produce a larger signal in comparison with the No-Ext-SF simulations.

For another sensitivity simulation, focusing on the internal seeder-feeder process, we inhibit in-situ ice particles (defined
here where CLC > 0) from sedimenting and crossing the $-35°C$ isotherm (Fig. 1a). This simulation is referred to as the No-Int-
SF simulation. Ice particles are allowed to be lofted from warmer temperatures across the $-35°C$. We chose the temperature,
$T = -35°C$, to be consistent with Herbert et al. (2015) and Proske et al. (2021).

Also of interest to us is the impact of ice particle breakup through ice-graupel collisions on the seeder-feeder process
(e.g. Sullivan et al., 2018; Dedekind et al., 2021, 2023). The enhancement of ice particles through secondary ice production
processes, other than the Hallet-Mossop process, has not been studied in light of the seeder-feeder process. Dedekind et al.
(2021) showed that allowing for the occurrence of breakup reduces the ice crystal and snow particle sizes and hence their fall



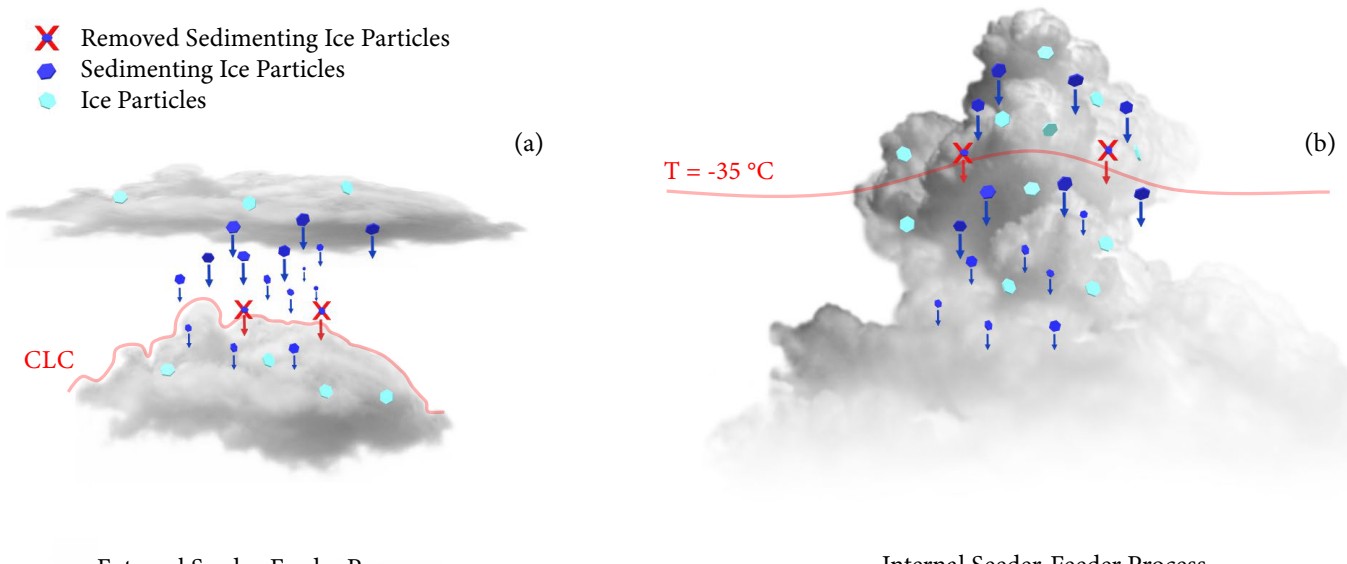

**Figure 1.** Schematic of the two seeder-feeder processes and the setup for the sensitivity simulations. The red line in both seeder-feeder cases is the level at which the seeding ice particles are removed. The simulations in which they are removed are referred to as **(a)** No-Ext-SF or **(a)** No-Int-SF (see Table 1).

speeds. This could therefore significantly reduce the sedimentation distances of ice particles in dry air (CLC = 0). In both the external and internal seeder-feeder inhibiting cases, additional simulations were conducted in which breakup occurs (No-Ext-SF_BR and No-Int-SF_BR simulations) and compared to a CNTL simulation in which breakup occurs (CNTL_BR simulation). Here, we used the same breakup parameters for BR as defined in Dedekind et al. (2021). See Table 1 for a summary of the

CNTL and sensitivity simulation setups.

The model and analysis domain roughly covers a region of $550 \times 650 \, \text{km}^2$ (44.5 to 49.5° N and 4 to 13° E) and $230 \times 235 \, \text{km}^2$ (45.5 to 47.5° N and 4.8 to 8° E), respectively, at a horizontal grid spacing of $1.1 \, \text{km} \times 1.1 \, \text{km}$. The analysis domain is used for all the analyses and figures in this study. A height-based hybrid smoothed level vertical coordinate system (Schär et al., 2002) with 80 levels was used and stretched from the surface to 22 km. Hourly initial and boundary conditions analysis data

at a horizontal resolution of $7 \, \text{km} \times 7 \, \text{km}$, supplied by MeteoSwiss, were used to force COSMO. For this study, we simulate a heavy precipitation event over eastern Switzerland. Our simulations are initiated at 10:00 UTC on May 18, 2016, and run until 22:00 UTC on that day. During this time a heavy precipitation event occurred from 17:30 to 20:45 UTC. Our analysis is split up into time periods from 13:30 to 17:00 UTC and 17:15 to 20:45 UTC (Fig. 2). During the initial period from 13:30 to 17:00 UTC many multi-layered clouds were developing, making it suitable to study the external seeder-feeder process. Later

during the day, 17:15 to 20:45 UTC, the clouds deepened significantly, stretching from close to the surface to about 10 km, which is appropriate for studying the internal seeder feeder process. Each simulation consists of 10 ensemble members. The





**Table 1.** Control and sensitivity simulations. Shown here are the processes that were either removed or included in each sensitivity simulation. When seeding ice particles are removed in the sensitivity simulations, no feeding of the lower-lying cloud occurs. These simulations are referred to as either the No External Seeder-Feeder Process (No-Ext-SF) or the No Internal Seeder-Feeder Process (No-Int-SF) simulation. Including collisional break or spherical ice particles (ice crystals and snow) are denoted by _BR and _Sph, respectively. All simulations include the rime splintering process.

|  | Control | No External Seeder-Feeder Process | No Internal Seeder-Feeder Process | Collisional Breakup | Spherical Ice Particles |
|---|---|---|---|---|---|
| CNTL | x |  |  |  |  |
| CNTL_BR | x |  |  | x |  |
| No-Ext-SF |  | x |  |  |  |
| No-Ext-SF_BR |  | x |  | x |  |
| No-Ext-SF_Sph |  | x |  |  | x |
| No-Int-SF |  |  | x |  |  |
| No-Int-SF_BR |  |  | x | x |  |

ensemble members are created by perturbing the initial temperature conditions at each grid point in the model domain with unbiased Gaussian noise at a zero mean and a standard deviation of 0.01 K (Selz and Craig, 2015; Keil et al., 2019).

The cloud area fraction computed by COSMO was compared to the DARDAR satellite data product by Proske (2020, their
Fig. 3.10) in an attempt to validate it. The output for one COSMO timestep at 12:30 UTC was compared to the overpass track of the satellite over eastern Switzerland at 12:32 UTC on May 18, 2016. Proske (2020) showed that COSMO was able to simulate the cloud layers observed in the DARDAR data remarkably well, with few exceptions where COSMO simulated more clouds than observed. Specifically, in regions close to the surface and beneath a thicker cloud layer, the discrepancy between COSMO and DARDAR can be attributed to either COSMO simulating cloud cover incorrectly or the radar signal from the satellite being attenuated when passing through the thicker top cloud layer. Considering that only a single time point from the model
was compared to the observations, no generalizations can be drawn from their conclusion.

In the analysis of multi-layered clouds (Section 3.1), we do not distinguish between the phase (ice, mixed or liquid) of the clouds in our simulations. As long as a multi-layered cloud was present within a column with a width and breadth of $1.1 \, \text{km} \times 1.1 \, \text{km}$, the layered cloud formed part of the analysis. For the seeder-feeder process to be recorded in the CNTL
simulation, two criteria need to be fulfilled. First, there needs to be a precipitation flux at the bottom of the seeder cloud and at the top of the feeder cloud. Second, when the precipitation flux into the feeder cloud occurs the ice particle mass mixing ratio has to be larger than $10^{-10} \, \text{g m}^{-3}$ otherwise the precipitation flux into the feeder cloud will be set to $0 \, \text{g m}^{-2} \, \text{s}^{-1}$ in order to avoid noise.



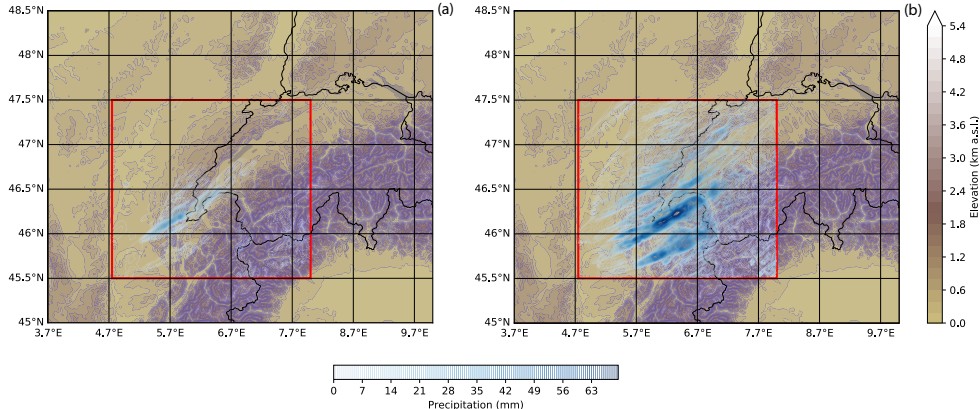

**Figure 2.** Overview of the model orography and the amount of precipitation. The large domain is the simulated domain and the red box is the domain used for the analysis. The accumulated precipitation is shown in **(a)** from 13:30 to 17:00 UTC and **(b)** from 17:15 to 20:45 UTC.

# 3 Results

## 3.1 Control simulation: multi-layered clouds vs seeder-feeder events

The multi-layered clouds and seeder-feeder events analyzed here are only for layered clouds that are separated by "clear" air, where $CLC = 0$ (see Fig. 1a). Figure 3 displays the frequency of occurrence of the total cloud tops, multi-layered clouds, and seeder-feeder situations. From 13:00 to 17:00 UTC, 48% of all clouds observed over the domain had another cloud below them while only a small fraction, 10%, of all layered clouds had ice particles precipitating from the seeder cloud into the feeder cloud. The 48% of layered cloud is comparable to the 31% reported by Proske et al. (2021) in their 10-year mean for several reasons. Firstly, here we only analyze a single case study in which a high precipitation event occurred which was selected because above-average multi-layer clouds were observed. Secondly, Proske et al. (2021) required that a pure ice cloud (cirrus) is above a lower-lying cloud, limiting the amount of possible seeder-feeder situations. The largest fraction of seeder-feeder occurrences (> 10% of all layered clouds) was found between 13:30 and 14:30 UTC. This corresponded to a mean vertically integrated precipitation flux into the feeder cloud of $\approx 1.3\,\mathrm{g\,m^{-2}\,s^{-1}}$ (Fig 3a and b). From 13:30 until 17:00 UTC the frequency of the external seeder-feeder process occurrences, layered clouds, and total cloud tops were reduced by 71%, 74% and 46%, respectively. The continuous development and deepening of the clouds were responsible for the strong reduction in layered clouds and the external seeder-feeder process as seen in the reduction of the distances between layered clouds (Fig 3b). The highest frequency of layered cloud occurrences, of $\approx 70\,000$ counts, had distances of 0.5 to 1.1 km compared to $\approx 2500$ occurrences with distances of 3 km (Fig. S1). Hall and Pruppacher (1976) and Proske et al. (2021) showed that ice crystals (plates, spheres or rosettes) typically can survive a fall of around 2 km (4 km at a very low occurrence frequency of less than 5%) depending on the relative humidity in the in-between air layers before they completely sublimate in the drier air layer. The ice particle survival fraction was calculated as the fraction of the ice particle precipitation flux rate at the top of the feeder



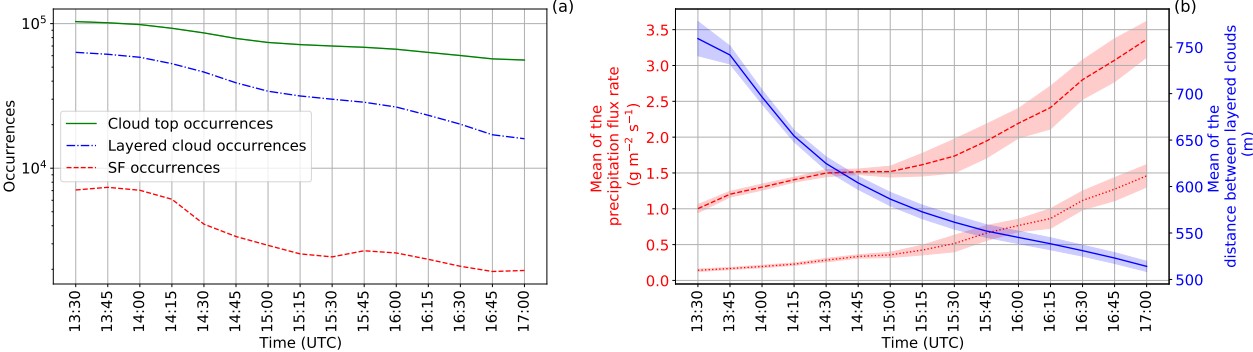

**Figure 3. (a)** The occurrence frequency of the total cloud tops, layered clouds (any occurrence of a cloud below another cloud) and seeder-feeder (SF) cases, and **(b)** the mean of the ice particle precipitation flux rate out of the seeder cloud (red dashed line) and into the feeder cloud (red dotted line) and distance between layered clouds in the CNTL simulation from 13:30 to 17:00 UTC. The shaded areas signify the 95% confidence interval of the control simulation. The 230 km×235 km analysis domain at a 1.1 km × 1.1 km horizontal resolution consists of about 45 000 columns.

cloud divided by the ice particle precipitation flux rate at the bottom of the seeder cloud. In this study, we show that there are

rare cases in which ice particles seeded a lower laying cloud that was between 2.5 km and 4.7 km below the seeder cloud (Fig. 4a and b). In some of these cases, the ice particle survival fraction was high, showing little loss in the ice particle precipitation flux rate. In general, the total precipitation flux summed over the analysis domain from 13:30 to 17:00 UTC at the bottom of the seeder clouds and at the top of the feeder clouds was 190 and 78 kg m$^{-2}$ s$^{-1}$, respectively. Therefore, in the seeder-feeder situations, approximately 60% of the ice particle mass was lost due to sublimation or melting of ice particles during

sedimentation. The high retention of the ice particle mass between the cloud layers is mainly because of the high occurrence of cloud layers being closer together than 1 km (see Fig. 4b). Proske et al. (2021) analyzed 2210 DARDAR-CLOUD satellite tracks set over Switzerland and showed the highest occurrences of seeder-feeder situations (which corresponds to our definition of layered clouds and is comparable to the blue histogram in 4a.) between 2 and 6 km (the dark green line in Fig. 4c). Our results suggest that shallower cloud layers compared to the 10-year climatology of Proske et al. (2021) and the high ice particle

retention between the cloud layers may have contributed to the heavy precipitation event.

      Next, we analyze the two time periods, 13:30 to 17:00 UTC and 17:15 to 20:45 UTC, which distinguish the external from the internal seeder-feeder process in the CNTL simulation. Fig. 5 provides an overview of the mass mixing ratios of the six hydrometeors in the CNTL simulation. Here, the temporal mean of the vertically integrated zonal mean mass mixing ratios is used for smoothing out the variability for each hydrometeor class. On 18 May 2016 from 13:30 to 17:00 UTC, the mean flow

was towards the north-east, causing the air mass to impinge on the mountain range between 5 to 6.7 °E and 45.8 to 46.5 °N (external seeder-feeder case, Figs. 1a and 2) promoting enhanced hydrometeor growth (Fig. 5a and b). The cloud (QC), rain (QR), snow (QS) and graupel (QG) mass mixing ratios all peaked between 45.8 to 46.25 °N where the total ice (QI + QS + QG + QH) and total liquid water (QC and QR) paths reached maximum values of 0.68 and 0.28 kg m$^{-2}$, respectively. All



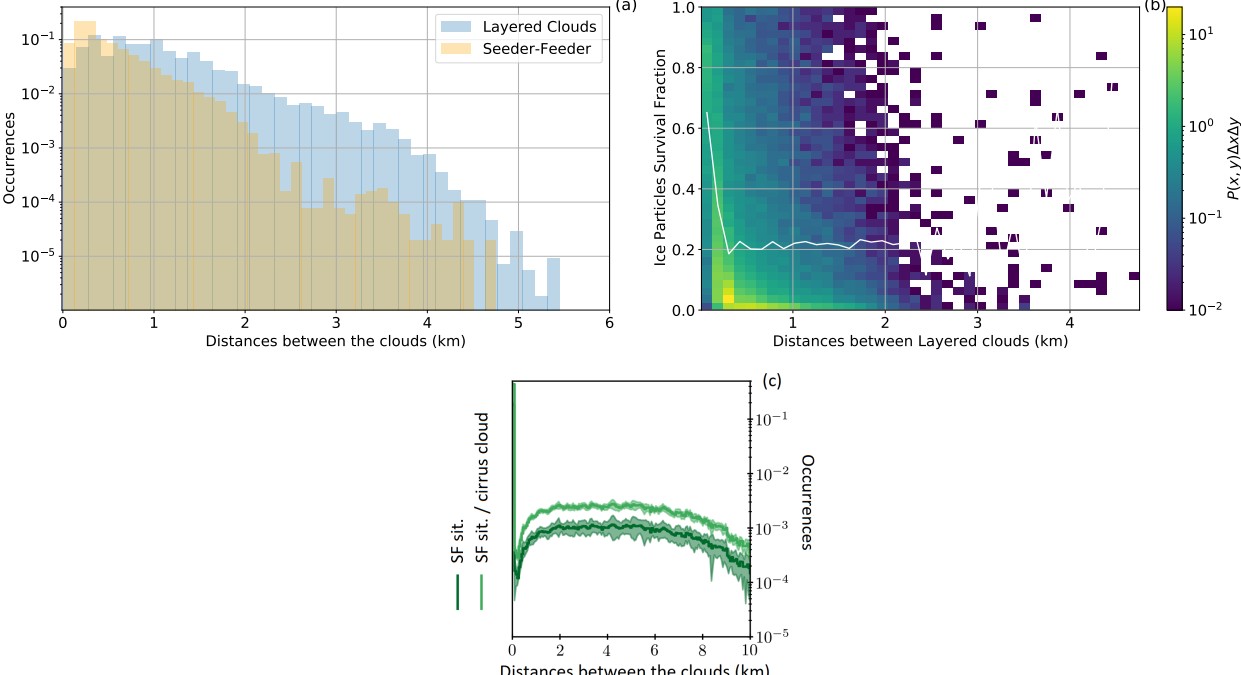

**Figure 4. (a)** The occurrence frequency of layered clouds and seeder-feeder situations plotted against the vertical distance between the clouds and **(b)** the joint probability density function multiplied by bin area ($P(x, y)\Delta x \Delta y$) for ice particle survival ($x$) between cloud layers ($y$) of varying distances from 13:30 to 17:00 UTC. The white line is the $50^{th}$ percentile as a function of the distance between layered clouds. The 230 km×235 km analysis domain at a 1.1 km×1.1 km horizontal resolution gives about 45 000 columns. 550 000 layered clouds and 55 000 seeder-feeder occurrences were observed during the time period. **(c)** The occurrence of seeder-feeder situations (SF sit. which is comparable to the layered clouds, the blue histogram, in panel **(a)**) against the vertical distance between clouds derived from 2210 radar lidar (DARDAR) satellite tracks between 2006 and 2017 (adapted from Fig. 3 by Proske et al., 2021).

the hydrometeors have a small variability between the ensemble members, except for the rain and graupel mass mixing ratios
with larger variabilities of 0.03 kg m⁻². The ice crystal mass mixing ratio exhibits the least variability between the ensemble
members and over latitude. The small variability in the ice crystal mass mixing ratio over latitude is most likely due to the fast
conversion processes from ice crystals to snow and graupel. Ice crystals can either aggregate to form snow or can interact with
cloud droplets and raindrops to form graupel or hail, granted that the riming rate is larger than the depositional growth rate
(Seifert and Beheng, 2006). In this case, ice crystals, in the vicinity of cloud droplets, are converted to graupel implying that
riming is the dominant growth process (also seen in Fig. S2).

In the evening, from 17:15 to 20:45 UTC, the precipitation increased significantly over a larger area with the precipitation
maximum being shifted toward the east. The increased precipitation rate is a result of the formation of deeper clouds coinciding
with fewer layered clouds (Figs. 1b and 3a). The increase in all hydrometeor mass mixing ratios, at least by a factor of two, is
associated with the deeper clouds (Fig. 5c and d). Due to the orographic forcing, the zonal and temporal mean of the cloud, rain



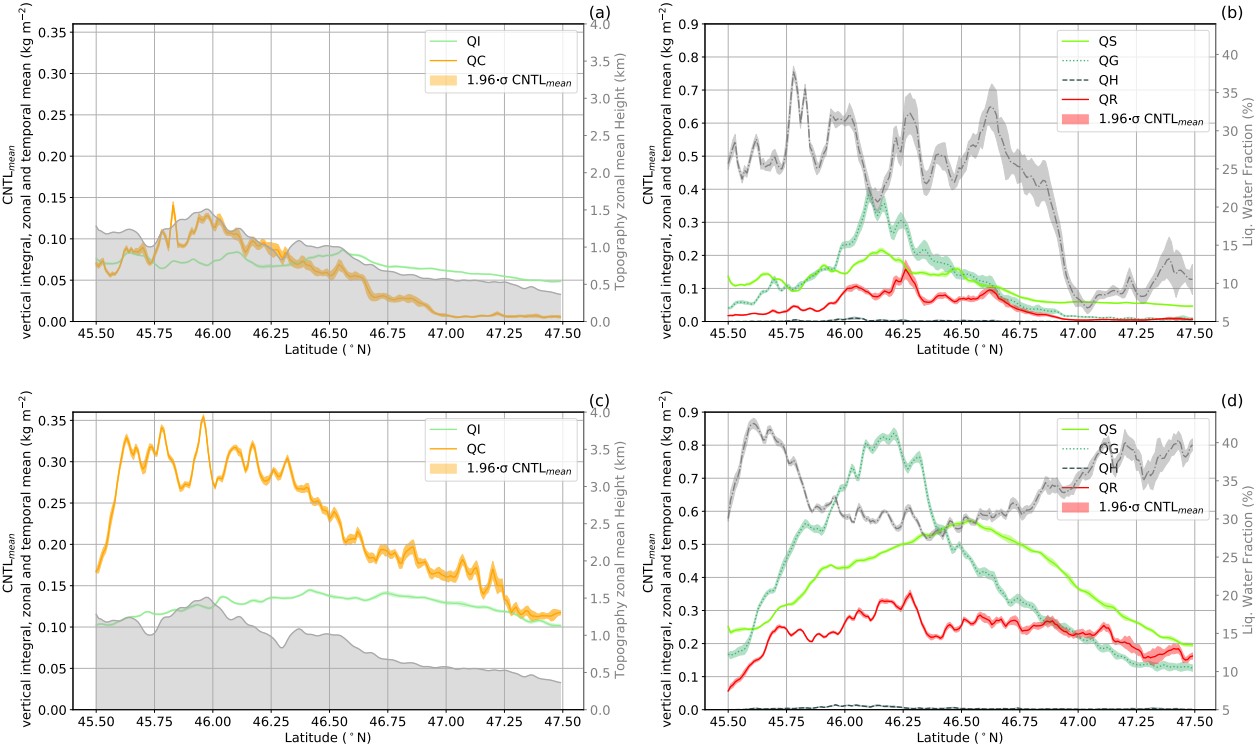

**Figure 5.** Hydrometeor mass mixing ratios for the CNTL simulations for **(a, b)** the external seeder-feeder (13:30 to 17:00 UTC) and, **(c, d)** the internal seeder-feeder processes (17:15 to 20:45 UTC). Shown is the temporal mean of the vertically integrated zonal means for **(a, c)** ice crystals, cloud droplets and the mean topography height (grey), and **(b, d)** snow, graupel, hail, rain drops and the liquid water fraction (grey). The liquid water fraction is calculated for temperatures $0 \geq T \geq -35\,°C$ to determine if the cloud is in a mixed-phase state. The shaded areas signify the 95% confidence interval of the control simulation.

and graupel mass mixing ratios peak between 45.8 to 46.25 °N. The ice and liquid water paths were higher, 1.4 and 0.7 kg m$^{-2}$ respectively, and consistent with a better-developed cloud that had twice as much water as compared to the cloud content from 13:30 to 17:00 UTC. During these two time periods, the clouds maintained a mixed-phase state with liquid water fractions ([(QC + QR)/(QC + QR + QI + QS + QG + QH)]·100) above 20% and ice particle growth by riming that extended throughout the cloud at $0 \geq T \geq -35\,°C$ (Figs. 5, S2a and S3a).

**3.2  External Seeder-Feeder**

**3.2.1  Changes in the cloud condensate**

To understand the impact of the seeder-feeder process on the cloud microphysics and surface precipitation, we inhibited seeding ice particles from precipitating into a feeder cloud from a seeder cloud. Figure S4a shows the vertically integrated, zonal and





temporal mean mass mixing ratios for all cloud condensate of the CNTL simulation which peaks at 0.82 kg m$^{-2}$ at 46.1 °N and

decreases to 0.15 kg m$^{-2}$ at 47.0 °N. The percentage change of the mass mixing ratio of all cloud condensate of the No-Ext-SF simulation from the CNTL simulation varies between −6.5% and 1.8% with an average reduction of 2.0% over the domain (45.5 °N and 47.5 °N). 31% of the deviations are outside of the CNTL simulation variability (outside of the 95% confidence interval) and are therefore significant (Fig. S5 and Table 2).

### 3.2.2  Changes in riming and depositional growth rates

Riming and depositional growth processes depend on the availability of ice particles. Reduced seeding ice particles can cause a reduction in these growth processes in the feeder cloud. Figures S5a and c illustrate that reducing the seeds causes a maximum reduction of 17% (3.1%) and an average reduction of 3.9% (0.55%) of the riming rate (depositional growth rate) (Table 2). Over the domain, 21% of the changes in the riming rate from the CNTL simulation variability were significant whereas only 6.5% of the changes in the depositional growth rate were significant. The reduction in the riming rate in the No-Ext-SF simulation

occurred mostly between 2 and 4 km in altitude, where cloud liquid was most abundant. In this region of the cloud a significant reduction in the riming rate of up to 10 % occurred, corresponding to 10$^{-4}$ kg m$^{-3}$ s$^{-1}$, because of the reduced ice particle mass mixing ratio in the No-Ext-SF simulation. Via reduced riming, removing seeding particles significantly impacts graupel formation (Fig. S6). To further illustrate the importance of the different growth processes in the seeder-feeder situations, specifically during the spring and summer seasons, we first calculated the difference between each sensitivity simulation (No-

Ext-SF and No-Ext-SF_Sph) and the CNTL simulation and likewise between the No-Ext-SF_BR simulation and the CNTL_BR simulation. After we normalized the precipitation and the growth rates, we calculated the correlation coefficients. In this case, the most significant growth process responsible for the seeder-feeder process was the riming rate with a correlation coefficient of 0.37 as shown in the No-Ext-SF simulation (Fig. 6a). The correlation coefficients between the depositional growth rate with surface precipitation were relatively weak, less than 0.13, and mainly insignificant (Fig. 6b). This suggests that when the

seeder-feeder process occurred in our case, the external ice particles' growth through deposition or aggregation (not shown here) did not contribute significantly to surface precipitation while riming did.

### 3.2.3  Changes in the surface precipitation

The precipitation change in the No-Ext-SF simulation compared to the CNTL simulation's variability over the domain was over 45%, especially south of 46.5 °N. Between 45.5 °N and 46.8 °N the No-Ext-SF simulation had at most 27 % and on average

8.5 % less surface precipitation which demonstrates the impact of seeding ice particles on precipitation formation (Fig. S7a and Table 2). It has to be considered that the precipitation difference could originate partly from removing the seeding particles at the feeder cloud top assuming the seeding particles would have survived the fall in dry air (without growing in the feeder cloud) until reaching the ground. If they had survived the fall, then the reduction in precipitation would originate merely from the removed seeding particles. To analyze this further, the lowest 10th and 50th percentiles of the distances of the feeder cloud

top to the surface were calculated. The 10$^{th}$ percentile shows the rare cases in which the feeder cloud tops were the closest to the surface representing higher probabilities for ice particles to survive a fall. The 10th and 50th percentile distances were





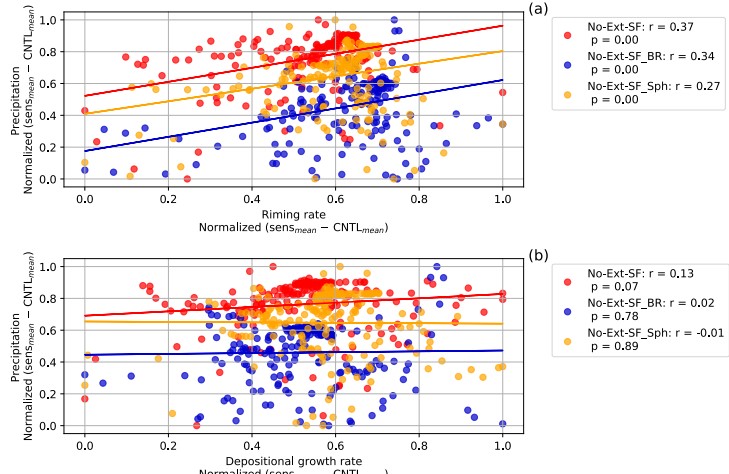

**Figure 6.** Correlations for the external seeder-feeder growth processes of **(a)** the riming rate and **(b)** the depositional growth rate of ice particles where $0 \geq T \geq -35\,^\circ$C with the surface precipitation rate. One point represents the vertically integrated, zonal and temporal mean of the growth rates correlated with the zonal and temporal mean of the precipitation rate at a latitude point from 13:30 to 17:00 UTC. Differences were calculated between the sensitivity (No-Ext-SF and No-Ext-SF_Sph) and the CNTL simulation ensemble means, and the No-Ext-SF_BR and CNTL_BR simulation ensemble means and normalized between 0 and 1. A linear least-squares regression for the growth processes and the surface precipitation is shown by the colored lines.

1.3 km and 2.3 km above the surface, respectively. Figure 4b illustrate that the seeding particles rarely sediment further than 2.5 km in dry air and also that the joint probability of ice particle survival over distances of 2 km was substantially reduced. Therefore, it is likely that only a small fraction of the seeding particles that were removed at the feeder cloud top would have contributed to the total reduction of surface precipitation and the majority of the reduction was a result of the missing external seeder feeder process.

### 3.2.4 Changes in the ice particle shape

We further analyzed whether the shape of ice particles impacts the external seeder-feeder process by assuming spherical crystals in the No-Ext-SF_Sph simulation. The No-Ext-SF_Sph simulation behaves similarly to the No-Ext-SF simulation, except that the snow mass mixing ratio was also significantly reduced compared to the CNTL simulation (Fig. S5a). The spherical snow has a faster fall velocity than the snow, represented as "mixed aggregates", in the CNTL simulation. Even though spherical snow has a larger ventilation coefficient that causes faster sublimation rates, more spherical snow reached the feeder cloud. Vassel et al. (2019) and Proske et al. (2021) showed similar results in calculating the sublimation rate and the survival of spherical particles when they sediment between cloud layers. The percentage loss of the mass mixing ratio of all cloud condensate was reduced by up to 7.9% and on average by 3.1% (Fig. S5a) when assuming spherical crystals. Surprisingly in our results, the additionally removed snow mass mixing ratio did not cause a substantially stronger decrease in the riming rates compared to





the No-Ext-SF simulation between 45.5 °N and 46.8 °N (Fig. S6a). There was a greater reduction in the depositional growth rate, especially between 2 and 4 km which was also apparent in the vertical integral (Figs. S2f and S6c). In general, the seeding particles in the No-Ext-SF_Sph simulation impacted the surface precipitation; however, the impact was not substantially larger
than in the No-Ext-SF simulation considering the reduction in all cloud condensate mass mixing ratios (Fig. S5a).

### 3.2.5  Including ice-graupel collisions

Also of interest was the multiplication of ice particles through the breakup of ice crystals and snow upon colliding with graupel in the context of the seeder-feeder process (No-Ext-SF_BR). Comparing CNTL_BR to CNTL across the zonal mean, the impact of ice-graupel collisions was clearly visible (Fig. S4). As the graupel mass mixing ratio increased, snow-graupel
collisions were promoted resulting in a decrease in the snow mass mixing ratio. The increase in the total ice water path (QI + QS + QG + QH) caused a reduction in the total liquid water path (QC + QR), which can be attributed to the significantly higher depositional growth rates (Fig. S6d). Due to the huge depositional growth rates below 6 km, there was an intensification in latent heat release and, consequently, in updraft velocity (Fig. S8). The ice particles were either lofted to higher regions in the cloud or sedimented at slower velocities against the strong updraft velocities, reducing the amount of precipitation
significantly. In turn, the liquid water fraction was reduced between 4 to 11 % compared to the CNTL simulation (Figs. 5b and S9b). Adding ice-graupel collisions led to larger changes in the liquid water fraction and riming rate from the seeder feeder process (comparing No-Ext-SF_BR to CNTL_BR). Inhibiting the more abundant smaller ice particles from falling into the feeder cloud yielded more profound positive and negative differences in both growth processes (riming and deposition), but this is not any more significant than the difference between the No-Ext-SF_Sph simulation and the CNTL simulations.
In the early stages of the development of MPCs during which breakup of ice particles occurred, the dominant growth process contributing to surface precipitation was riming with a correlation coefficient of 0.34 as compared to deposition and aggregation with correlation coefficients of 0.02 and −0.12 (not shown here), respectively, which are not statistically significant (Fig. 6a and b). During a field campaign at Davos in the Swiss Alps, Ramelli et al. (2021) measured cloud properties from an airborne balloon. They found evidence for a large fraction of rimed ice particles including graupel during a seeder-feeder event, which
confirms the importance of the riming process in the feeder cloud. Albeit aggregation rates can increase substantially (e.g., Dedekind et al., 2021; Georgakaki et al., 2022) as a direct consequence of having more ice particles from SIP, this did not have a significant impact on surface precipitation in our simulations (Fig. 6a and b). Georgakaki et al. (2022) found the aggregation of snow to be a major driver of secondary ice formation in external seeder-feeder events. In their case, they considered ice-ice collisions (Phillips et al., 2017) which did not require rimed particles for fragmentation to occur and, therefore, could lead to
SIP in an ice cloud (seeder cloud). Here, we are only including ice-graupel collisions which require riming to form graupel. Further development is required to describe the collisional tendencies between unrimed ice particles within the Seifert and Beheng (2006) two-moment cloud microphysics scheme.





### 3.3 Internal Seeder-Feeder

#### 3.3.1 Changes in the cloud condensate

The internal seeder-feeder sensitivity simulations were analyzed from 17:15 to 20:45 UTC during which the clouds were well-developed and extended to about 10 km. The impact of natural cloud seeding is studied by removing the seeding ice particles before they sediment into the mixed-phase part of the cloud at $T > -35\,^\circ$C (internal seeder-feeder process, Fig. 1b). The removal of ice particles at $\approx 7$ km in the No-Int-SF simulation led to a maximum and average reduction of 8.3% and 5.8%, respectively, of the mass mixing ratios of all cloud condensate (Fig. S10a and Table 2). These percentage changes were significant over 98%

of the domain (Fig. S10a).

#### 3.3.2 Changes in the riming and depositional growth rates

Over the domain, the impact of removing seeding particles on the riming rate (depositional growth rate) was more (less) pronounced with a maximum and average reduction of 5.6% (3.4%) and 1.4% (0.77%), respectively, compared to the CNTL simulation (Fig. S11a, c and Table 2). Between 46.00 °N and 46.25 °N, the depositional growth rate was significantly enhanced,

reaching a percentage change of 3.0%, compared to the CNTL simulation (Fig. S11c). Less weight of the hydrometeors to be carried by the updraft velocity likely contributed to the stronger updraft velocities in the No-Int-SF simulation (Fig. S12). The enhanced updraft velocity in conjunction with an enhanced liquid water fraction (Fig. S13a), as a result, promoted higher supersaturations with respect to ice, enhancing the growth of ice particles by vapor deposition (Fig. S3d). However, the reduced average depositional growth rate and a significantly enhanced maximum and average liquid water fraction of 8.9% and 6.0%,

respectively, show that the removal of seeding ice particles slowed the glaciation in the mixed-phase part of the cloud over the domain.

#### 3.3.3 Changes in surface precipitation

Albeit the depositional growth rate had a similar fraction of significant latitudinal grid points over the domain (34% of latitude grid points) compared to the riming rate, the depositional growth rate did not have an impact on surface precipitation whereas

the riming rate showed to be the dominant precipitation forming process with a correlation coefficient of 0.5 with surface precipitation (Fig. 7a). The No-Int-SF simulation showed a maximum and average reduction of the surface precipitation of 7.5% and 3.0%, respectively, which was significant over 69% of the latitude grid points compared to the CNTL simulation (Fig. S14a).

Here we have shown that inhibiting the internal seeder-feeder effect deamplifies cloud glaciation. Because of the reduction

in cloud condensate the ice particle growth processes are slowed down causing a reduction in surface precipitation.





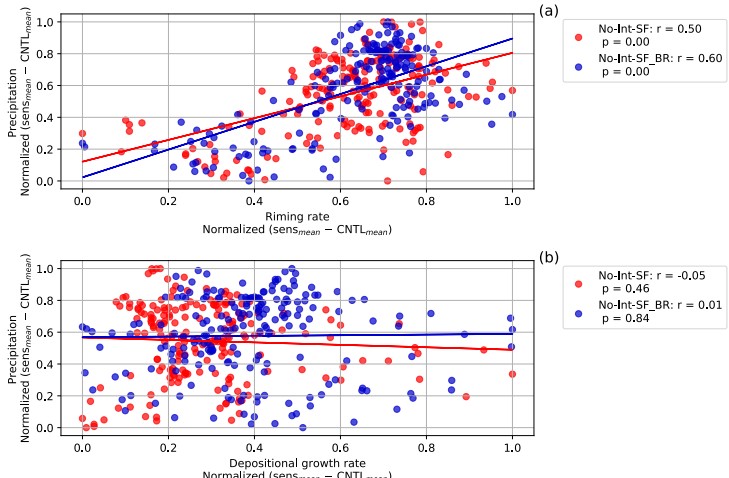

**Figure 7.** As Figure 6, but for the internal seeder-feeder growth processes from 17:15 to 20:45 UTC.

### 3.3.4 Including ice-graupel collisions

Including ice-graupel breakup into the CNTL simulation (CNTL_BR) significantly changes the cloud characteristics compared to the CNTL simulation. The significant increase (decrease) in the cloud condensate (liquid water fraction) compared to the CNTL simulation results from the huge amount of smaller ice particles (similar to Dedekind et al., 2021) feeding on the
available liquid water through the WBF process when the supersaturation is between the saturation with respect to water and ice (Figs. 10a, b, and S15). The smaller ice particles have smaller sedimentation velocities and cannot fall against the stronger updrafts and therefore have longer residence times and enhanced growth by deposition. At altitudes above 5 km, the cloud was glaciated in the CNTL_BR simulation (Fig. S3e), which is also manifested in a 3 to 8% reduction in liquid water fraction (Fig. S15), slowing down the precipitation formation process.

Removing seeding ice particles in the No-Int-SF_BR simulation that originated from regions in the cloud where $T < -35\,°C$, decreased the mass mixing ratio of all cloud condensate on average by 3.4% (significant over 98% of the domain), resulting in a higher average liquid water fraction of 6.0% compared to the CNTL_BR simulation (Figs. S9b, S13b and Table 2). The fewer ice particles in the No-Int-SF_BR simulation cause a significant decrease in the depositional growth rate between 5 and 7 km (Fig. S3d). However, here too, the change in the deposition rate is not correlated with the change in precipitation.

Having fewer particles that eventually sediment to lower altitudes meant that the riming rate was also significantly less between 45.8 °N to 46.6 °N and below 3 km (Figs. S3 and S11b) which strongly correlated with the reduction in precipitation (Fig. 7a). Interestingly, the differences in surface precipitation between the No-Int-SF and CNTL simulations and the No-Int-SF_BR and CNTL_BR simulations due to inhibiting the internal seeder-feeder process are very similar. Rutledge and Hobbs (1983b) showed that in the absence of seeding, the hydrometeors remained in the liquid phase which suggests that seeding events were





characterized by ice particle growth through the WBF process. In accordance, here we show that removing ice particles, which sediment from cirrus levels, deamplifies cloud glaciation, and reduces surface precipitation.

## 4    Conclusions

Simulations of a strong precipitation event associated with a high occurrence of multi-layered clouds on 18 May 2016 in the Swiss Alps were carried out with the non-hydrostatic limited-area atmospheric model, COSMO. A two-moment cloud
microphysics scheme describing the evolution of the number density and mass mixing ratio of six hydrometeor species was used within COSMO to investigate the seeder-feeder process and its impact on precipitation formation. The seeder-feeder process was subdivided into two processes and analyzed in separate simulations, inhibiting the external (No-Ext-SF) and internal seeder-feeder processes (No-Int-SF), and compared against a control (CNTL) simulation.

At first, we analyzed the control simulations to describe the frequency of occurrence of multi-layered clouds against the external
seeder-feeder processes, and our findings can be summarized as follows:

- From 13:30 to 17:00 UTC, 47.6% of all clouds, derived from only masked cloudy regions, were categorized as multi-layered clouds which are comparable to what Proske et al. (2021) identified from their 10-year satellite data analysis over Switzerland. 10.3% of the multi-layered clouds seeded a lower-lying feeder cloud with ice particles.

- When the seeder-feeder process occurred, 60% of the ice particle mass was lost between cloud layers due to sublimation
or melting. The high retention of ice particle mass between cloud layers results from the high occurrence of short (less than 0.9 km) distances between multi-layered clouds.

Thus our simulations confirm that also in COSMO, seeder-feeder simulations are frequent and that a considerable amount of ice particles survives the sedimentation into a lower cloud.

Second, we analyzed the sensitivity simulations for the external seeder-feeder process, from 13:30 to 17:00 UTC, and the
impact that the seeding particles had on the feeder cloud. Table 2 provides an overview of the percentage changes of the sensitivity simulations for the analyzed fields compared to the CNTL and CNTL_BR simulations.

- Inhibiting the external seeder-feeder process significantly reduced the amount of cloud condensate south of 46.25 °N. That is because the reduced cloud condensate reduced the riming rate over the domain, which was the most prominent growth process for precipitation formation.

- Including breakup in the external seeder-feeder process did not change the cloud content difference with respect to CNTL. In this case, the riming process was also the dominant growth process of ice particles that led to surface precipitation. Higher amounts of the total cloud condensate were retained within the clouds causing less surface precipitation in the No-Ext-SF_BR simulation compared to the No-Ext-SF simulation.

These simulations highlight that the ice particles from the seeder cloud not only influence the feeder cloud, but also subse-
quent precipitation formation. The findings on the role of deposition and riming in the seeder feeder process have implications





**Table 2.** Average and maximum change in surface precipitation, mass mixing ratio of all cloud condensate, liquid water fraction, riming rate and depositional growth rate of the No-Ext-SF and No-Int-SF simulations compared to the CNTL and the No-Ext-SF_BR and No-Int-SF_BR simulations compared to the CNTL_BR simulation (numbers in brackets).

|  | No External Seeder-Feeder Process | | No Internal Seeder-Feeder Process | |
|---|---|---|---|---|
|  | Average change | Maximum change | Average change | Maximum change |
| Surface precipitation | −8.5 % (−8.8 %) | −27 % (−30 %) | −3.0 % (−3.4 %) | −7.5 % (−7.9 %) |
| MMR cloud condensate | −2.0 % (−2.2 %) | −6.5 % (−9.1 %) | −5.8 % (−5.8 %) | −8.4 % (−8.9 %) |
| Liq. water fraction | −0.70 % ** (−2.2 % **) | −8.9 % ** (−19 % **) | 6.0 % (6.0 %) | 8.9 % (10 %) |
| Riming rate | −3.9 % (−5.4 %) | −17 % (−22 %) | −1.4 % (−2.6 %) | −5.6 % (−9.4 %) |
| Depositional growth rate | −0.55 % ** (−0.52 % **) | −3.1 % ** (−5.5 % **) | −0.77 % (−0.72 %) | −3.4 % (−3.2 %) |

** Less than 10% of grid points over the zonal mean showed significant changes

for the process' significance over Switzerland: Across the Swiss Alps, the liquid water fraction for wintertime mixed-phase cloud is less than ∼20% (Henneberg et al., 2017; Dedekind et al., 2021, 2023). Dedekind et al. (2021) found that simulated mixed-phase clouds have a low liquid water fraction compared to observed mixed-phase clouds. These clouds favor the depositional growth, during which the WBF process (Korolev, 2007) is the dominant precipitation process. However, this process does not show to impact the seeder-feeder process. During summer, the enhanced liquid water fraction of ∼35% favors riming as the dominant growth formation process which is significant for the seeder-feeder process.

Third, we analyzed the sensitivity simulations for the internal seeder-feeder process, from 17:15 to 20:45 UTC, in which seeding ice particles are inhibited from sedimenting from cirrus ($T < −35°C$) levels within the same cloud.

- By removing the seeding ice particles, significantly less cloud condensate seeded the lower part of the cloud from cirrus levels, which reduced the ice particle number densities resulting in a higher liquid water fraction. Inhibiting the internal seeder-feeder process reduced the surface precipitation significantly, but not by the same magnitude that the cloud condensate decreased (Table 2) which is opposite to the external seeder-feeder case.

- In both the No-Int-SF and No-Int-SF_BR simulations, removing seeding ice particles generally resulted in a decrease in the deposition rate at $0 \geq T \geq −35 °C$ pointing towards a deamplification in cloud glaciation.

In general, we found the external seeder-feeder process to be more important than the internal seeder-feeder process in terms of percentage changes of riming and surface precipitation. The simulations with the COSMO model allowed us to investigate the process mechanism and effect in detail. They indicate that natural cloud seeding impacts both feeder cloud glaciation and surface precipitation. Together with the frequency estimate from Proske et al. (2021), our results make the case for the seeder-feeder process being both a frequent and impactful process with the need for further study. In particular, both observational and model studies with a larger domain and sample size are needed to elucidate the global significance of the process.

Our study has implications for model development as well since it highlights the importance of simple ice sedimentation parameterisation. The process may lie at the root of model cloud microphysics behaviour. For example, the global climate



model ECHAM-HAM has been shown to be insensitive to heterogeneous freezing (Hoose et al., 2008; Dietlicher, 2018; Dietlicher et al., 2019; Villanueva et al., 2021; Ickes et al., 2022, 2023; Proske et al., 2023), a process which in turn is believed
to be important in the atmosphere (J. Murray et al., 2012; Kanji et al., 2017b). One hypothesis for the insensitivity is that the sedimentation of ice crystals is so strong in the model that it supplies ample ice crystals to lower-lying clouds, leaving no room for heterogeneous freezing to act as an ice-triggering process. In turn, other microphysical processes may impact the importance of the seeder-feeder process. For example, ice-ice collisions (which are currently not included in the COSMO model due to a lack of observational data) may result in the formation of ice particles of smaller diameters in the seeder cloud.
In this case, the fractured seeding ice particles will fall shorter distances and may not reach the feeder cloud, rendering the seeder-feeder process less important. In terms of model development, the question of how to represent ice sedimentation is further linked to the cloud microphysics scheme choice. The 2 moment scheme in particular introduces an artificial divide between ice crystals and snowflakes, where snow reaches the ground via sedimentation within one timestep. Schemes like the P3 CMP scheme (Morrison and Milbrandt, 2015; Dietlicher et al., 2018) overcome this divide and treat all ice sedimentation
explicitly. Our study adds to the evidence that ice sedimentation as a process deserves more consideration in model investigation and development, showing that it influences cloud interactions and precipitation formation via the seeder-feeder process. In addition, it also adds an approach of how to constrain the sedimentation process' formulation: by investigating the seeder-feeder process as demonstrated here, with a combination of satellite and modeling studies, we may constrain the magnitude and influence that ice crystal sedimentation has in the atmosphere and should have in weather and climate models.



*Data availability.* The COSMO model output used for our analysis is available at https://doi.org/10.5281/zenodo.7637600 and the software to analyze the data can be found at https://doi.org/10.5281/zenodo.7637604.



## 4.1

*Author contributions.* ZD conducted the simulations and analyzed the results. ZD and UP were the main authors of the paper. UP, DN, UL and SF contributed to the design of the study and the analysis of the results. All authors contributed to the writing of the study.

*Competing interests.* The authors declare that they have no conflict of interest.

*Acknowledgements.* All simulations were performed with the Consortium for Small-scale Modeling (COSMO) model. The simulations were performed and are stored at the Swiss National Supercomputing Center (CSCS) under project s1009 and s903. ZD and UL acknowledge funding from the Swiss National Science Foundation (SNSF) grant number 200021_175824. UL, UP and DN acknowledge funding from the European Commission, H2020 Research Infrastructures (FORCeS) grant number 821205.



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
