# Peer review of "Simulating the seeder-feeder impacts on cloud ice and precipitation over the Alps"

_EGUsphere, 2023_

## Referee Comment (RC1)

Review of: Simulating the seeder-feeder impacts on cloud ice and precipitation over the Alps

by Zane Dedekind, Ulrike Proske, Sylvaine Ferrachat, Ulrike Lohmann, and David Neubauer

Manuscript egusphere-2023-874

General comments

In this paper, the impact of so-called seeder-feeder events on cloud properties and surface precipitation are investigated in the frame of model sensitivity studies. From an upper seeder cloud ice particles fall out reaching the lower mixed-phase feeder cloud where they may change the cloud microphysics and, eventually, precipitation. To quantify these effects model simulations were performed with and without seeding effects and, additionally, with and without secondary ice formation, and with and without the assumption of spherical ice particles instead of snow aggregates or ice crystals. The simulations were performed with the 3D cloud model system COSMO combined with a two-moment microphysical scheme. From their results, the authors deduced the importance of seeder-feeder effects, not only on the microphysical properties of the feeder cloud such as liquid and ice water content, updraft and latent heat release, but also on in-cloud processes such as riming rate and depositional growth rate, and, finally, on the development of precipitation. Furthermore, they come out with suggestions for future model studies.

The paper represents a substantial contribution to the research in this field. The authors used a real case for their simulations that took place in Switzerland where the topography over the Alps favors frequent occurrence of natural cloud seeding. The scientific approach and the applied methods are valid, and the results are discussed in an appropriate way. The paper is very well written and well-structured.

To my opinion, the paper is worth publication after minor revisions.

Specific comments:

Page 2 line 39: What exactly do you mean by accretion? The deposition of water vapor? Please clarify in the text.

Page 3 line 56 and others: Describe explicitly that the seeder cloud is the upper cloud and the feeder cloud the lower one (as I understand it).

Page 4 line 99: Explain the terms cloud area fraction and the abbreviation CLC.

Page 7 lines 158 to 160 and Figure 3: How did you calculate these values of 48%, 10%, and 31%? Are they mean values over the entire time period? Please specify this in the text and add the range of the values.

Page 7 line 164: Instead of > 10%, give a more specified value.

Page 8, Figure 3: What exactly do you mean with "occurrence frequency of total cloud tops"? Occurrence of any clouds?

Page 12, title of section 3.2.4: I think the title is somehow misleading as not the effect of seeding on the ice particle shape is studied but the effect of the ice particle shape on the seeder-feeder process. I suggest to change the title accordingly.

Page 13 last paragraph: It is not completely clear to me at the end how breakup after ice-graupel collisions affects precipitation. First you discuss about riming, then about aggregates, finally about ice-ice collisions that are not included in your model. Please rework this paragraph to make your results and conclusions clearer to the reader.

Page 16 line 365: What do you mean by "cloud condensate difference"?

Page 17 line 373: I cannot not agree with this statement. Could you give a reference that growth via the deposition of water vapor is generally the dominant process and not riming? You write later that during

summer the enhanced liquid water fraction favors riming as the dominant growth process. Why not in winter? Please overwork these phrases.

Technical corrections:

Page 1 line 18: Please remove the reference in the Abstract.

Page 4 line 119: Maybe change "further distances" into "longer distances".

Page 4 line 114: I think this must be Fig. 1b.

Page 5, caption of Figure 1: I think "… referred to as … or (**b**) No-Int-SF …" should be correct.

Page 7, caption of Figure 2: Reformulate the sentence "The large *domain* is the simulated *domain* … "

Page 7 line 158: Please correct the time from 13:00 to 13:30.

Page 7 line 165: please correct Fig**s.** 3a and b, line 168: Fig**.** 3b.

Page 8, caption of Figure 3: I suggest to move the last sentence into the text where you describe the model setup.

Page 8, Figure 4b, title of x axis: change Layered into layered.

Page 8 line 174: I suggest to reformulate the sentence into: We note from Figs. 4a and b that there are rare cases in which … that was more than 2.5 km below the seeder cloud.

Page 8 line 176: It is not clear which cases you mean by "some of these cases". Please rewrite this sentence.

Page 8 line 183: … blue histogram in Fig. 4a) … (remove the point before the second bracket).

Page 9, caption of Figure 4: I suggest to move the sentences "The 230 km×235 km analysis domain at a 1.1 km×1.1 km horizontal resolution gives about 45 000 columns. 550 000 layered clouds and 55 000 seeder-feeder occurrences were observed during the time period." into the text, see comment above.

Last sentence in figure caption: change into: " … against the vertical distance between clouds derived from Proske et al., 2021. (The other information is already given in the text.)

Page 11 line 232: please change "riming rate" into "riming". A growth *process* is riming, not the riming rate.

Page 12 line 247: Figure 4b illustrate**s** …

Page 12 line 251: seeder**-**feeder process

Page 13 line 274: change into: "… at slower velocities against the strong updraft." (omit the last "velocities").

Page 13 line 276: seeder**-**feeder

Page 13 line 280: I would suggest to write here mixed-phase clouds instead of MPCs because for some readers this abbreviation is not well-known although it was explained some pages before.

Page 13 line 283 and 287: Fig**s**. 6a and b

Page 13 line 286: I think the abbreviation of secondary ice production (SIP) is not explained in the text.

Page 14 line 305: Omit the word "percentage".

Page 15 line 325: Please introduce the abbreviation of the Wegener-Bergeron-Findeisen process (WBF) because not every reader may be familiar with this, for example in the Introduction where you mention this process.

Page 16 line 341: Remove the comma after "glaciation".

Page 16, section 4: I suggest to rename this section in "Summary and Conclusions" because the main part is a summary of your results.

Page 16 line 344:  Remove the comma before "COSMO".

Page 16 lines 351 and 359:  I suggest to omit the time data because they are not relevant in this context.

Page 16 lines 355/356:  I suggest to change: " … the high occurrence of short distances smaller than 0.9 km between multi-layered clouds."

Page 16 line 357:  I suggest to change: " … confirm that seeder-feeder events are frequent …"

Page 16 lines 360/361: Please reformulate this sentence. Table 2 provides the changes of precipitation and so on, but not the changes of the sensitivity simulations. You could write "changes found in sensitivity simulations". See comment below.

Page 16 lines 362-364: Please rework this phrase, there must be a mistake. Additionally, the statement "south of 46.25°N" should be omitted here.

Page 16 line 370: seeder-feeder

Page 17, Table 2: This table belongs into the section containing the results and not in the Conclusion section. I suggest to move the table together with the above commented phrase into section 3 where it is firstly mentioned.

Page 17, caption of Table 2: Please correct: "Average and maximum change**s**  ( … ) riming rate**,** and depositional growth rate  (…)  compared to the CNTL_BR simulation**s** …

Page 17 line 372: mixed-phase cloud**s**

Page 17 line 373: What is meant by "these clouds" – observed clouds or mixed-phase clouds? Please mention it clearly in the text. The formulation that the WBF process is the dominant precipitation process is not completely correct, the WBF process promotes precipitation formation. Please rewrite this phrase.

Page 17 line 377:  I suggest to omit the time data.

Page 17 line 382:  Omit "(Table 2)".

Page 18 line 401: I suggest to start a new paragraph before you discuss the microphysical schemes.

Page 18 line 402: Please change into " The **two-**moment scheme …"

Figures S2 and S3, captions: I suggest to change "Cross-section of the growth **rate** of ice particles".

" … for the ensemble means**,** respectively."

It seems that Figure S3 is not mentioned in the paper.

---

## Author Comment (AC1)

**Simulating the seeder-feeder impacts on cloud ice and precipitation over the Alps**

Zane Dedekind[1,2,*], Ulrike Proske[2,*], Sylvaine Ferrachat[2], Ulrike Lohmann[2], and David Neubauer[2]

[1]Department of Earth, Ocean, and Atmospheric Sciences, University of British Columbia, Earth Sciences Building, 2207 Main Mall, Vancouver, BC, V6T 1Z4, Canada
[2]Institute of Atmospheric and Climate Science, ETH Zurich, Switzerland
[*]These authors contributed equally to this work.

**Correspondence:** Zane Dedekind (zane.dedekind@ubc.ca) and David Neubauer (david.neubauer@env.ethz.ch)

We sincerely thank the Reviewers for the constructive feedback. The suggestions and comments considerably improved the quality of the manuscript.

Below we present a detailed response with the reviewer comments in black, our responses in blue and additions to the manuscript in blue italics.

**Reviewer 1**

**Specific comments**

1. Page 2 line 39: What exactly do you mean by accretion? The deposition of water vapor? Please clarify in the text.

   Accretion is the growth of ice particles by deposition and/or riming. We adapted the sentence:

   *Here, an orographically formed cloud acts as the feeder cloud, in which the ice particles that sediment from the seeder cloud into the feeder cloud grow by accretion (deposition and/or riming) or aggregation.*

2. Page 3 line 56 and others: Describe explicitly that the seeder cloud is the upper cloud and the feeder cloud the lower one (as I understand it).

   Yes, that is correct. Instead of mentioning it in the text, we have added more detail to the caption of Figure 1.

   *In panel (**a**) the seeder and feeder clouds are the higher and lower clouds, respectively. In panel (**b**) the seeder and the feeder clouds are separated at a temperature of $-35\,°C$, with the seeder cloud laying above the feeder cloud, respectively.*

3. Page 4 line 99: Explain the terms cloud area fraction and the abbreviation CLC.

   To calculate the cloud cover (CLC), the in-cloud water mixing ratio is first calculated. The in-cloud water mixing ratio is a function of the mass mixing ratios for cloud droplets (QC) and ice crystals (QI) from the subgrid-scale and convective cloud schemes. Whenever $QC > 0\,g/kg$ and/or $QI > 10^{-7}\,g/kg$ then CLC is calculated. Therefore, $CLC = 0$ is defined as regions outside of the cloud. We added the following description to the manuscript:

*The in-cloud regions are determined by calculating the in-cloud water mixing ratios as a function of the mass mixing ratios for cloud droplets (QC) and ice crystals (QI) from the subgrid-scale and convective cloud schemes. Whenever $QC > 0\,g\,kg^{-1}$ and/or $QI > 10^{-7}\,g\,kg^{-1}$, then the cloud cover (CLC, a full 3D function) is larger than 0. Therefore, CLC = 0 is defined as the out-of-cloud regions, or in this case, the region between the cloud layers.*

4. Page 7 lines 158 to 160 and Figure 3: How did you calculate these values of 48%, 10%, and 31%? Are they mean values over the entire time period? Please specify this in the text and add the range of the values.

   In Fig. 3, we mention that our analysis is over the domain which consists of about 55 000 columns (each grid point over the domain was analysed). The 48% layered cloud fraction and 10% seeder-feeder case fraction are the cumulative occurrences over the time period from 13:30 to 17:00 UTC. For example, 48% of the time when any cloud was observed in a column, there was another cloud below between 13:30 and 17:00 UTC.Likewise, 10 % of the time when layered clouds were observed in a column, the seeder-feeder process also occurred.

5. Page 7 line 164: Instead of > 10%, give a more specified value.

   The manuscript was updated with more specific values:

   *between 10% and 12% of all layered clouds.*

6. Page 8, Figure 3: What exactly do you mean with "occurrence frequency of total cloud tops"? Occurrence of any clouds?

   Yes, that is correct.

7. Page 12, title of section 3.2.4: I think the title is somehow misleading as not the effect of seeding on the ice particle shape is studied but the effect of the ice particle shape on the seeder-feeder process. I suggest to change the title accordingly.

   We changed the title to:

   *Impact of the ice particle shape on the seeder-feeder process*

8. Page 13 last paragraph: It is not completely clear to me at the end how breakup after ice-graupel collisions affects precipitation. First you discuss about riming, then about aggregates, finally about ice-ice collisions that are not included in your model. Please rework this paragraph to make your results and conclusions clearer to the reader.

   To elucidate the impact of breakup on precipitation, we included the following description in the Summary and Conclusions section:

   *The effect of breakup on precipitation could be ambiguous. If breakup occurs in the vicinity of supercooled cloud droplets, breakup should enhance precipitation via the WBF and riming processes. Conversely, in the absence of supercooled cloud droplets, breakup is expected to yield smaller ice crystals, thereby increasing the likelihood of sublimation below the cloud base.*

   Yes, the suggested flow of the paragraph is correct. We first discuss the impact of ice-graupel collisions on microphysical rates. Secondly, we compared our analysis against other studies from Ramelli et al. (2021) and Georgakaki et al. (2021).

However, a key difference when comparing aggregation rates between our work here and the Georgakaki et al. (2021) work is that the parameterizations, as described in the manuscript, are fundamentally different.

9. Page 16 line 365: What do you mean by "cloud condensate difference"?

Thank you for asking this. We removed this sentence because it is not clear what was meant here.

10. Page 17 line 373: I cannot not agree with this statement. Could you give a reference that growth via the deposition of water vapor is generally the dominant process and not riming? You write later that during summer the enhanced liquid water fraction favors riming as the dominant growth process. Why not in winter? Please overwork these phrases.

In Henneberg et al. (2017), Dedekind (2021) and Georgakaki et al. (2022) vertical profiles of growth rates are shown for winter-time MPCs. In all these studies the depositional growth rate was larger than the riming rate. These references are given in the manuscript. We reworked the paragraph to make our statements more clear.

*Because of the lower liquid fraction, the interaction between liquid water and ice particles (the riming process) is less efficient than during summertime. In wintertime mixed-phase clouds the ice crystals mainly grow via deposition, to which the WBF process (Korolev, 2007) largely contributes (Henneberg et al., 2017; Dedekind et al., 2021; Georgakaki et al., 2022). During summertime, the significance of the depositional growth process appears to be of minor importance in influencing the seeder-feeder process.*

**Technical corrections**

Thank you for your careful reading and the below suggestions. Most of them we have incorporated exactly as you indicated. If not, we only inserted our response below.

1. Page 1 line 18: Please remove the reference in the Abstract.

2. Page 4 line 109: Maybe change "further distances" into "longer distances".

3. Page 4 line 114: I think this must be Fig. 1b.

4. Page 5, caption of Figure 1: I think "... referred to as ... or (b) No-Int-SF ..." should be correct.

5. Page 7, caption of Figure 2: Reformulate the sentence "The large domain is the simulated domain ... "

*The maps show the whole simulated domain while the red box is the domain used for the analysis.*

6. Page 7 line 158: Please correct the time from 13:00 to 13:30.

7. Page 7 line 165: please correct Figs. 3a and b, line 168: Fig. 3b.

8. Page 8, caption of Figure 3: I suggest to move the last sentence into the text where you describe the model setup.

We have removed the sentence from the caption and amended the model domain description in the Methods section to include the information:

*The model and analysis domain roughly covers a region of $550 \times 650\,km^2$ (44.5 to 49.5° N and 4 to 13° E) and $230 \times 235\,km^2$ (45.5 to 47.5° N and 4.8 to 8° E), respectively, at a horizontal grid spacing of $1.1\,km \times 1.1\,km$. The analysis domain consists of about 55 000 columns and is used for all the analyses and figures in this study.*

9. Page 8, Figure 4b, title of x axis: change Layered into layered.

10. Page 8 line 174: I suggest to reformulate the sentence into: We note from Figs. 4a and b that there are rare cases in which . . . that was more than 2.5 km below the seeder cloud.

   We changed the sentence to:

   *We note from Figs. 4a and b that there are rare cases in which ice particles seeded a lower laying cloud that was between 2.5 km and 4.7 km below the seeder cloud.*

11. Page 8 line 176: It is not clear which cases you mean by "some of these cases". Please rewrite this sentence.

   We specified the cases to be the rare long-distance cases from the sentence before.

   *In some of these long-distance cases, the ice particle survival fraction was high, showing little loss in the ice particle precipitation flux rate.*

12. Page 8 line 183: . . . blue histogram in Fig. 4a) . . . (remove the point before the second bracket).

13. Page 9, caption of Figure 4: I suggest to move the sentences "The 230 km×235 km analysis domain at a 1.1 km×1.1 km horizontal resolution gives about 45 000 columns. 550 000 layered clouds and 55 000 seeder-feeder occurrences were observed during the time period." into the text, see comment above. Last sentence in figure caption: change into: " . . . against the vertical distance between clouds derived from Proske et al., 2021. (The other information is already given in the text.)"

   We have decided to keep the information in the caption for easy reference. We added the additional information in the "Model setup" section as requested above.

14. Page 11 line 232: please change "riming rate" into "riming". A growth process is riming, not the riming rate.

   Thanks for pointing us to this error in formulation.

   *In this case, the most significant growth process responsible for the seeder-feeder process was riming. The riming rate has a correlation coefficient of 0.37 as shown in the No-Ext-SF simulation (Fig. 6a).*

15. Page 12 line 247: Figure 4b illustrates . . .

16. Page 12 line 251: seeder-feeder process

17. Page 13 line 274: change into: ". . . at slower velocities against the strong updraft." (omit the last "velocities").

18. Page 13 line 276: seeder-feeder

19. Page 13 line 280: I would suggest to write here mixed-phase clouds instead of MPCs because for some readers this abbreviation is not well-known although it was explained some pages before.

20. Page 13 line 283 and 287: Figs. 6a and b

21. Page 13 line 286: I think the abbreviation of secondary ice production (SIP) is not explained in the text.

    *You are correct. We have replaced the use of the abbreviation with the full term.*

22. Page 14 line 305: Omit the word "percentage".

23. Page 15 line 325: Please introduce the abbreviation of the Wegener-Bergeron-Findeisen process (WBF) because not every reader may be familiar with this, for example in the Introduction where you mention this process.

    *We have included the explanation of the abbreviation in the introduction.*

    *Ice crystals can grow by riming or vapor deposition (including the rapid growth via the Wegener-Bergeron-Findeisen (WBF) process, where ice crystals grow at the expense of cloud droplets when the air is subsaturated with respect to liquid water but supersaturated with respect to ice (Wegener, 1911; Bergeron, 1935; Findeisen, 1938)).*

24. Page 16 line 341: Remove the comma after "glaciation".

25. Page 16, section 4: I suggest to rename this section to "Summary and Conclusions" because the main part is a summary of your results.

26. Page 16 line 344: Remove the comma before "COSMO".

27. Page 16 lines 351 and 359: I suggest omitting the time data because they are not relevant in this context.

    *We have decided to keep this as part of our summary because it describes the larger context of multi-layered cloud and seeder-feeder events.*

28. Page 16 lines 355/356: I suggest changing: "... the high occurrence of short distances smaller than 0.9 km between multi-layered clouds."

29. Page 16 line 357: I suggest changing: "... confirm that seeder-feeder events are frequent..."

    *We appreciate your suggestion. However, we would like to keep the scope of our statement limited to COSMO, rather than extending it to other models. This is because different models have different approaches to treating sedimentation, and the survival of ice particles between cloud layers is crucial for accurately simulating the seeder-feeder process.*

30. Page 16 lines 360/361: Please reformulate this sentence. Table 2 provides the changes of precipitation and so on, but not the changes of the sensitivity simulations. You could write "changes found in sensitivity simulations". See comment below.

We reformulated the sentence:

*Table 2 provides an overview of the percentage changes of the sensitivity simulations compared to the CNTL and CNTL_BR simulations for the analyzed fields.*

31. Page 16 lines 362-364: Please rework this phrase; there might be a mistake.

    Yes, that was a mistake, sorry:

    *Inhibiting the external seeder-feeder process significantly reduced the amount of precipitation south of 46.25 °N.*

32. Page 16 line 370: seeder-feeder

33. Page 17, Table 2: This table belongs to the section containing the results and not in the Conclusion section. I suggest moving the table together with the above-commented phrase into section 3 where it is first mentioned.

    We moved Table 2 and the corresponding phrase into Section 3.

34. Page 17, caption of Table 2: Please correct: "Average and maximum changes (...) riming rate, and depositional growth rate (...) compared to the CNTL_BR simulations..."

35. Page 17 line 372: mixed-phase clouds

36. Page 17 line 373: What is meant by "these clouds" – observed clouds or mixed-phase clouds? Please mention it clearly in the text. The formulation that the WBF process is the dominant precipitation process is not completely correct; the WBF process promotes precipitation formation. Please rewrite this phrase.

    Thank you again for highlighting our confusing statement. You are correct and we addressed it in the Specific Comments Section point 10.

37. Page 17 line 377: I suggest omitting the time data.

38. Page 17 line 382: Omit "(Table 2)"

39. Page 18 line 401: I suggest starting a new paragraph before you discuss the microphysical schemes.

40. Page 18 line 402: Please change into "The two-moment scheme..."

41. Figures S2 and S3, captions: I suggest changing "Cross-section of the growth rate of ice particles." "...for the ensemble means, respectively." It seems that Figure S3 is not mentioned in the paper.

    We corrected the captions. Figure S3 is mentioned in five places within Section 3.

**Reviewer 2**

1. Experiment Design and Simulation Evaluation: The authors should provide more details on the experiment design and simulation evaluation. How about the evolution of simulated clouds compared to the observed? How about the comparison in precipitation? When to remove the ice particles in the sensitivity experiments? Only during the two time periods analyzed in this study?

   These comments are important, specifically how well the model can simulate clouds. The significance of these lies in the model's ability to accurately simulate layered clouds, which is crucial for the external seeder-feeder process. Our efforts involved comparing the model's performance with DARDAR observations, as documented in the study by Proske (2020). This reference is also cited in our manuscript. Notably, south of 45 °S, the model tends to simulate thicker feeder clouds than observed, potentially leading to an overestimation of riming process (Fig. 1 here). However, due to the limitations of polar-orbiting satellites with infrequent revisit times, we caution against generalizing this comparison. For robust statistical validation, we recommend conducting a follow-up study that includes multiple case studies or exploring climate simulations, which can be compared to multiple satellite passes for further model assessment.

   We added the following description of the ice particles removal in the sensitivity studies:
   *Between 13:00 and 17:00 UTC, ice particles were removed between cloud layers for the external seeder-feeder process (detailed in the next paragraph). For the internal seeder-feeder process the in-situ ice particles were removed between 17:15 and 20:45 UTC.*

   We simulated these two time periods because during these two periods, we could observe clear instances of both the external and internal seeder-feeder processes. As mentioned earlier, it would be beneficial to expand this study and explore climate simulations in conjuction with satellite observations for statistical significance.

2. Uncertainties in Microphysics Scheme: The study is mainly based on the simulations of a real case, so the authors should conduct further analysis on how the uncertainties of the microphysics scheme, especially the parameterization of some microphysical processes such as riming and depositional growth influence the seeder-feeder process. Is it possible to give some average vertical profiles of process rates in different simulations to show their changes clearly?

   The vertical profiles depicted here in Figures 2 and 3 represent the averaged riming and depositional growth rates over the domain, specifically within in-cloud regions. The shaded areas correspond to the uncertainties arising from ensemble members in each simulation. Notably, at an altitude of approximately 2.25 km, the riming rate in the No-Ext_SF simulation exhibits a significant reduction compared to the CNTL simulation. Interestingly, this reduction is not observed in the depositional growth rates. This finding aligns with our analysis, emphasizing the importance of riming in the external seeder-feeder process.

3. Line 33: "Wegener-Bergeron-Findeisen" -> "Wegener-Bergeron-Findeisen (WBF)"

   This agrees with point 23 from reviewer 1 and has been implemented there.

[Figure]

(a)

(b)

**Figure 1.** Proske (2020, Fig. 3.10): Direct comparison between simulation ctrl02 (output from 12:30 pm) and the satellite data (track enters the domain at 12:32 pm). a) Vertical crosssection of satellite and COSMO simulation data along the position of the satellite track shown in b), as a function of temperature. Blue colors display the cloud cover simulated by COSMO, orange marks the cloud base and green the cloud top as seen by the DARDAR satellite product (variable frac_cov ). Where the satellite track encompasses several pixels in width, the sum of frac_cov over these pixels is displayed. When this is larger than 1, the cloud base/top is marked in a deeper shade of orange/green. In b) the grey shades mark the topography in the COSMO simulation domain.

[Figure]

**Figure 2.** The in-cloud riming and depositional growth rates for all the external seeder-feeder simulations (13:30 to 17:00 UTC) for **(a, b)**, respectively, averaged over the whole domain. The shaded areas signify the 95% confidence interval of each simulation. The black line is the mean temperature.

[Figure]

**Figure 3.** The same as Fig. 2, but for the internal seeder-feeder process (17:15 to 20:45 UTC)

4. Line 39: "sedimenting ice particles from the feeder cloud" or "from the seeder cloud"?

   We corrected the sentence to"...sedimenting ice particles from the seeder cloud..."

5. Line 99: "CLC" or "CAF"? How does the model define "cloud area fraction"? based on cloud mixing ratio?

   The name cloud area fraction and cloud cover is defined by CLC in the model. Both these variables are calculated in the same way. Therefore, we modified the text from cloud area fraction to cloud cover. For the specific definition see point 3 (Specific comments) from reviewer 1.

6. Line 114: "Fig. 1a" -> "Fig. 1b"

   Yes, see point 3 (Technical corrections) from reviewer 1.

7. Figure 2: Colorbar is not clear.

8. Line 158: "13:00" -> "13:30"?

   Yes, see point 6 from reviewer 1.

9. Figure 3: What's "occurrence frequency of the total cloud tops"?

   We reworded the it. See Reviewer 1, Specific Comments, point 6. It is the frequency of any clouds occurring in a grid box summed across the entire domain and time-period.

10. Lines 234-236: The authors analyzed correlation coefficient here. I would say higher coefficient does not necessarily mean higher contribution, especially if the authors calculated correlation coefficient using simultaneous time series. The

ice growth through deposition can contribute to the later surface precipitation. The authors should rewrite the related analysis and draw conclusions carefully.

We agree that relying solely on correlations is inadequate. Ice growth via deposition can potentially influence subsequent surface precipitation. However, our findings, as depicted in Figure S2e (also summarized in Table 2), indicate that even if this hypothesis holds true, the deposition rate across the cloud depth remains largely insignificant and would not significantly impact surface precipitation. This observation aligns with the low correlation coefficient of 0.13 (Fig. 6b). Our No-EXT_SF simulation, conducted over a duration of 3.5 hours with 15-minute output intervals, should be sufficiently long to detect any substantial differences. Figures 6 and 7 further validate our conclusions. In contrast, the riming rate in the No-Ext_SF simulation significantly diverges from that in the CNTL simulation near the melting layer just above 2 km in altitude (as shown in Figure 2 (here) and Fig. S2b), corroborating the findings presented in Table 2. Given the substantial changes in riming and precipitation (Table 2) and the correlation coefficient of 0.37, it is plausible that riming is the dominant process in precipitation formation, particularly within the context of our case study.

11. Section 3.2.5: Does the microphysics scheme include rime splintering process? How does this process rate change in sensitivity experiments?

Rime splintering is included and used in the model. Unfortunately, we do not have output for the rime splintering rate. Figures 2a and 3a (here) show the vertical profile for the mean temperature over the domain. Between 265 and 270 K, the region in which rime splintering occurs, there is a clear indication of enhanced riming rates. However, from the mean vertical profiles, there is no evidence of significant differences in the rime splintering process rates between the sensitivity simulations.

12. Line 286: "SIP" -> "secondary ice production (SIP)"

Yes, see point 21 from reviewer 1.

13. Line 326: "Figs. 10a, b" -> "Figs. S10a, b"?

Thank you for pointing us to this typo, which we corrected.

**References**

Dedekind, Z.: The Impact of the Ice Phase on Orographic Mixed-phase Clouds and Surface Precipitation in the Swiss Alps, Doctoral Thesis, ETH Zurich, https://doi.org/10.3929/ethz-b-000511939, 2021.

Dedekind, Z., Lauber, A., Ferrachat, S., and Lohmann, U.: Sensitivity of precipitation formation to secondary ice production in winter orographic mixed-phase clouds, Atmospheric Chemistry and Physics, 21, 15 115–15 134, https://doi.org/10.5194/acp-21-15115-2021, 2021.

Georgakaki, P., Bougiatioti, A., Wieder, J., Mignani, C., Ramelli, F., Kanji, Z. A., Henneberger, J., Hervo, M., Berne, A., Lohmann, U., and Nenes, A.: On the drivers of droplet variability in alpine mixed-phase clouds, Atmospheric Chemistry and Physics, 21, 10 993–11 012, https://doi.org/10.5194/acp-21-10993-2021, 2021.

Georgakaki, P., Sotiropoulou, G., Vignon, , Billault-Roux, A.-C., Berne, A., and Nenes, A.: Secondary ice production processes in wintertime alpine mixed-phase clouds, Atmospheric Chemistry and Physics, 22, 1965–1988, https://doi.org/10.5194/acp-22-1965-2022, publisher: Copernicus GmbH, 2022.

Henneberg, O., Henneberger, J., and Lohmann, U.: Formation and Development of Orographic Mixed-Phase Clouds, Journal of the Atmospheric Sciences, 74, 3703–3724, https://doi.org/10.1175/JAS-D-16-0348.1, 2017.

Korolev, A.: Limitations of the Wegener–Bergeron–Findeisen Mechanism in the Evolution of Mixed-Phase Clouds, Journal of the Atmospheric Sciences, 64, 3372–3375, https://doi.org/10.1175/JAS4035.1, 2007.

Proske, U.: Estimation of the importance of natural cloud seeding, MSc Thesis, https://doi.org/10.3929/ethz-b-000477328, 2020.

Ramelli, F., Henneberger, J., David, R. O., Lauber, A., Pasquier, J. T., Wieder, J., Bühl, J., Seifert, P., Engelmann, R., Hervo, M., and Lohmann, U.: Influence of low-level blocking and turbulence on the microphysics of a mixed-phase cloud in an inner-Alpine valley, Atmospheric Chemistry and Physics, 21, 5151–5172, https://doi.org/10.5194/acp-21-5151-2021, 2021.